# Perception of rural adolescents and parents regarding child marriage: Findings of a community-based cross-sectional study in Bangladesh

Md. Ziaul Islam[1]*, S. M. Sharf-ul-alam[2], Farjana Farha[3], Nargis Sultana[3], Ananya Adhya[4], Sharmin Farjana[5]

1 Professor and Head, Department of Community Medicine, National Institute of Preventive and Social Medicine, Dhaka, Bangladesh, 2 Assistant Chief Statistical Officer, National Institute of Preventive and Social Medicine, Dhaka, Bangladesh, 3 Research Fellow, National Institute of Preventive and Social Medicine, Dhaka, Bangladesh, 4 Lecturer, Community Medicine, Khulna Medical College, Khulna, Bangladesh, 5 Medical Officer (OSD), Department of Reproductive Endocrinology and Infertility, Bangabandhu Sheikh Mujib Medical University, Shahbag, Dhaka, Bangladesh

* dr.ziaul.islam@gmail.com

## Abstract

### Background

Child marriage is a blazing global issue influenced by gender inequity, poverty, social norms, and insecurity. It has disastrous social, economic, and health-related consequences. Bangladesh ranks among the top ten countries in terms of child marriage, and rural girls are the most vulnerable group to it. Our study aimed to assess and compare the perception of rural adolescents and parents regarding child marriage, as well as focusing on the key factors governing such perception of the participants.

### Methods

This community-based cross-sectional study randomly selected four unions of Dhamrai upazila (Sub-district) under the Dhaka district as study areas. The study recruited 1030 participants (515 adolescents and 515 parents) from 515 rural households using a systematic random sampling method and specific selection criteria. Data enumerators collected data through face-to-face interviews using a pretested semi-structured questionnaire and obtained informed written consent from parents and assent from adolescents. The study assessed perception related to child marriage by 25 statements formulated based on expert opinions. These statements were further organized into four domains- social, economic, psychological, and environmental through thematic analysis. We categorized the perception as "support" and "do not support" child marriage using a scoring system ranging from 0 to 25, based on the responses to 25 statements by a two-level Likert scale. Participants who obtained scores of ≥80% were considered not to support child marriage.

**Data availability statement:** All data underlying the findings described in the manuscript are uploaded anonymously as Supporting Information.

**Funding:** The author(s) received no specific funding for this work.

**Competing interests:** Authors declare no competing interests of any type.

## Results

The current study revealed that 25.8% of adolescents and 27.0% of parents supported child marriage. Participants lacking formal schooling were 3 times (AOR 3.00; 95% CI 1.27–7.09, p = 0.01) and participants with primary level education were 2.69 times (AOR 2.69; 95% CI 1.27–7.09, p = 0.01) more likely to support child marriage compared to participants with at least higher secondary level of education. Parents and adolescents did not differ significantly (p < 0.05) in total and domain-specific perception scores. However, substantial differences existed in various statements across the domains, and maximum disagreement was found in the environmental domain.

## Conclusion

Supportive perception toward child marriage still exists in rural Bangladesh, and it is more prevalent among people with less or no formal education. Despite exceptions, both groups had overall similar perceptions regarding child marriage. Exploration of factors favoring support for child marriage and designing educational as well as community-based interventions specific and suitable to adolescents and parents is crucial to improve perception regarding child marriage.

## Background

Child marriage is a burning issue affecting millions of children, especially girls, all over the world. The UN Convention on the Rights of the Child (CRC) and the Children's Act, 2013 of Bangladesh, defined a child as any person under the age of 18. Accordingly, the CRC and the United Nations High Commissioner for Human Rights define child marriage as the formal or informal marriage of any person under the age of 18 with or without consent [1,2]. This type of marriage is mostly forced marriage occurring as a result of deeply seated perceptions in patriarchal societies regarding the necessity of child marriage, fueled mainly by social norms, economic hardship, and an insecure environment for females [3].

Despite a significant reduction in the global burden of child marriage over the last decades, every day 39,000 and each year about 12 million girls are married off before the age of 18 years. Worldwide, more than 700 million women are alive at present who have become victims of child marriage [4,5]. The prevalence varies between communities across the globe, ranging from 1.8 to 90.85% [6]. Sub-Saharan Africa and South Asia regions have the highest rates of girls undergoing child marriage, 38.0% and 30.0%, respectively [7]. Child marriage is a cruel reality for millions of adolescent girls, especially those living in developing countries where every 1 in 7 girls is married before the age of 15 [3]. According to UNICEF, in developing countries during 2007–2017, 12.0% of girls were married before the age of 15 and 38.0% before the age of 18 years [1]. If this rate persists, an estimated 151 million child marriage cases will occur during 2021–2030 [8].

Southeast Asia shares about half of the global burden of child marriage [9]. The prevalence of child marriage among girls in India, Indonesia, and Nepal is 27.0%, 14.0%, and 40.0%, respectively [10]. Bangladesh has a long tradition of early marriage, especially for girls, and ranks among the top 10 countries in the world in terms of child marriage prevalence [11]. However, the rate of child marriage declined from 88.7% to 45.7% from 1951 to 2011 [3]. This reduction in prevalence reflects the effectiveness of various social and legal preventive measures. The Child Marriage Restraint Act, 2017 of Bangladesh prohibits marriage for Bangladeshi girls under the age of 18 years as they are not eligible to give consent for marriage. Despite the law, according to Bangladesh Demographic and Health Surveys (BDHS), 59.0% of women got married before the age of 18 and 22.0% before the age of 15 during 2017–18 [12]. The rate of child marriage is significantly higher in rural communities, where about 71.0% of women aged 20–24 years are married before 18 [13]. Kohno et al., reported that poor enforcement of existing laws has significantly impeded the efforts to reduce child marriage to the expected level in many developing countries [14].

Child marriage is recognized as a violation of human rights, as it engenders the health of adolescent girls as well as their children, disrupts education, limits economic opportunities, and denies the joy of childhood life [15]. Adolescent brides have less access to contraceptive services and minimal control over their conception time and interval, leading to early and multiple childbearing [16,17]. They are more vulnerable to unwanted pregnancy, unsafe abortion, and obstetric complications like obstetric fistulas, eclampsia, post-partum hemorrhage, etc., leading to high rates of mortality and morbidity [18,19]. Evidence suggests that pregnancy-related complications are the second most common cause of death among 15–19-year-old women [20]. The maternal mortality rates are 2 and 5 times higher among girls aged 15–19 and 10–14 years than in women aged 20–24 years [21]. Moreover, child marriage is associated with an increased risk of HIV/AIDS, cervical cancer, and poor mental health among adolescent mothers [22].

Child marriage affects both women and their children in various stages of development. Child marriage is associated with a higher risk of prematurity, intrauterine growth retardation, stillbirth, and under-5 mortality and morbidity [22–25]. Evidence suggests that adolescent mothers have a significantly higher risk of delivering low-birth-weight babies [26,27]. Furthermore, children of young mothers are more likely to develop malnutrition and anaemia [28]. A study conducted in Africa showed that children born to mothers under 18 have a 25% higher risk of developmental problems and a 29% higher risk of stunting in comparison to babies born to older mothers [29].

In addition, girls marrying before 18 years are more likely to be victims of school dropout, social isolation, familial maladjustment, and domestic violence by intimate partners [21]. They typically lack autonomy, have limited or absent peer networks, face mobility restrictions, and have poor access to mass media like TV, Radio, and Newspapers compared to boys or unmarried girls [1,26]. Early marriage creates a significant obstacle to continuing and completing the studies of adolescent girls, which reduces their chances of earning a decent livelihood. As a result, a vicious cycle of child marriage, low education, poverty, discrimination against girls, and ultimately increased prevalence of child marriage is completed, which is very difficult to tear apart [30].

Social norms, culture, and economic constraints are the main drivers for child marriage. Traditionally, in many societies, women are taught to marry and assume the responsibility of the family as early as possible [26]. Poor parents regard girls as a burden on the family, and marrying them off lessens their economic burden. Moreover, some families don't want to delay the marriage of girls beyond their teenage years due to the fear of a high dowry to be paid to the groom's family for older brides [3]. Evidence suggests that women who married before 18 years are more likely to live in poverty and have higher living expenses than those who married after 18 years [21,31]. Child marriage curtails education and better job opportunities for both girls and boys. According to the World Bank report, ending child marriage in Bangladesh could result in a 12.0% increase in the earnings and productivity of women [32].

Religion is a crucial factor in shaping social norms and culture, which influences social behavior as well as the perception of people [6,33]. Bangladesh is a Muslim-majority country where about 90.0% of people are followers of the religion, Islam [34]. According to the teachings of Islam, parents are the key persons liable for selecting suitable life partners

for their daughters. However, it is often misinterpreted and misused to justify the imposition of the wishes of parents on adolescent girls regarding the age of marriage without considering their physical and psychological condition [35]. A group of Muslims believe that child marriage is allowed in Islam, and they give the example of the Prophet Mohammad (SM) marrying a nine-year-old Aisha (RA). On the contrary, some Muslims hold different beliefs and are against child marriage. They consider physical and emotional maturity necessary for the marriage of a woman [4]. However, a systematic review conducted in Southeast Asian countries showed inconclusive evidence regarding the role of any particular religion behind the occurrence of child marriage [36].

Some countries and societies with high gender inequality (due to laws and customs that exclude girls from decision-making or economic and political rights) are more likely to have a high prevalence of child marriage [2,34]. Child marriage is deeply embedded within social norms in Bangladesh, especially in rural areas [25,37]. As soon as a girl starts menstruating, she is considered an adult and suitable for marriage. Families with unmarried older girls have to face harsh comments and negative attitudes from society. As a result, guardians feel consistent pressure from relatives and neighbor to marry off their adolescent girls [35]. Protecting girls from sexual harassment, premarital relationships, and illegal sexual interaction is another vital factor for child marriage. Agege et al reported that premarital conception is the most common cause of child marriage, and all these marriages were forced by the parents [38]. It is also perceived that early marriage secures the virginity of girls and protects them from premarital conception, which are highly demanding criteria for the marriage of a girl [3].

In patriarchal societies, fathers usually decide when a girl should marry and with whom to marry. However, in most cases, the opinion of the adolescent girl is ignored, and it is expected that they will cope with the marriage process. Unfortunately, most guardians are reluctant to take responsibility for the consequences of these early marriages [39]. The inferior position of women in family and society triggers child marriage even more, as neither mothers nor girls can protest or have the right to protest against the decision of the male guardians [26]. According to studies by UNFPA, the Ford Foundation, and Plan International, many adolescent girls sometimes willingly accept marriage as their fate since they have little understanding of other options in life and comply with marriage due to social or emotional pressure [34].

The rate of child marriage is considered as an indicator of a nation's development. Due to devastating consequences and negative impact, child marriage has taken a prominent place in Target 5.3 of the Sustainable Development Goals (SDGs), stating the elimination of all harmful practices against girls, women, and children by 2030 [26,40]. These actions invite accelerated preventive interventions against child marriage, and according to UNICEF, a 23% reduction in prevalence by 2030 is essential to achieve this goal [6]. Moreover, the prevention of child marriage would help to reduce population growth remarkably in developing countries [32]. In Bangladesh, the National Plan of Action (NPA) was adopted in 2018, targeting to end child marriages under 15 years and 18 years by 2021 and 2041 in Bangladesh [41]. Despite some improvements, the reduction of the child marriage rate is not satisfactory yet [42]. Understanding the perception of the concerned population groups, adolescents and parents, may help to find valuable insights regarding the factors behind child marriage.

To our knowledge, this study is the first of its kind in Bangladesh, identifying and comparing perceptions of child marriage between adolescents and parents. These two successive generations are directly involved in the occurrence of child marriage, and the most suitable population groups to give intervention for reducing this cruel practice in Bangladesh. The study will shed light on vulnerable groups of parents and adolescents with a poor perception of child marriage, which might propel them towards it. Identifying the difference in perception will help to understand the viewpoints of both groups, which would help in designing appropriate and specific communication strategies and interventions for rural parents and adolescents to improve their perception. The study findings will also add value to the existing child marriage prevention campaign in Bangladesh.

## Methods

### Study design, settings, and participants

We started the community-based cross-sectional study on the 1st of January and completed it by the 30th of June 2023. The study enrolled two groups of participants- adolescents and parents. Perceptions regarding different domains of child

marriage were categorized and compared between the two groups. We developed the research protocol and data collection instrument in January 2023. After that, we prepared the field for data collection in February 2023. The study recruited data enumerators and field supervisors for two months (from 1st March to 30th April 2023), and they were given comprehensive training on the study objectives and procedure, along with data collection. We used a multi-stage systematic random sampling technique to select study sites. Out of 8 divisions of Bangladesh, we considered the Dhaka division, and out of 13 districts of the division, one district (Dhaka) was randomly selected. Followed by, out of 5 upazilas of Dhaka district, one upazila (Dhamrai) was chosen randomly, and four out of 16 unions were selected randomly (Fig 1). The four unions of Dhamrai upazila (sub-district) of Dhaka district were Taltola, Chhoybaria, Borochondrail, and Chotochondrail. A union is the lowest administrative unit of the Government of the People's Republic of Bangladesh.

**Sample size and sampling**

The sample size was calculated using following formula [43]

$$n = \frac{z^2 \left[ \left\{ p_1 \left( 1 - p_1 \right) \right\} + \left\{ p_2 \left( 1 - p_2 \right) \right\} \right]}{d^2}$$

Here, $p_1$ = 25.8% and $p_2$ = 26.0% indicating the proportion of parents and adolescents with supportive perception regarding child marriage respectively [26], $d$ = 5% = 0.05 and z = 1.96 at 95% CI. Putting all the values in the equation and adding a 10% non-response rate, the calculated sample size was 1054. We attended 1087 households of four unions and identified 1054 adolescents and parents as eligible participants. We invited 1054 (527 adolescents and 527 parents) to participate in the study. Considering the unwillingness to participate (12, 1.1%), unavailability (8, 0.8%), and incomplete interviews (4, 0.4%), 1030 participants (515 adolescents and 515 parents) were enrolled in the study from 4 unions.

   To enroll study participants, we first approached the selected union administrative offices (Union Parishad). Each Union Parishad is governed by a Chairman, elected through direct voting of the permanent residents of that respective union. The Chairman serves as the public representative at the lowest tier of the Government in rural settings of Bangladesh [44]. Obtaining the household list from the Union Parishad, we gave a unique number to each house.

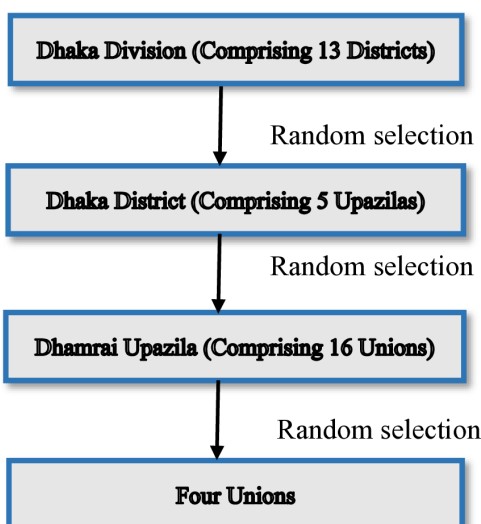

**Fig 1. Sampling technique for selecting four unions as study places.**

Based on the sample size, the number of households was allocated proportionately among four unions. We selected households from each union using a systematic random sampling technique following a calculated skip interval. From each selected household, one adolescent boy or girl (aged 12–19 years) and one of his/her parents were enrolled randomly as study participants. All the participants were permanent residents of those rural settings (unions). In case of the absence of any participant in a household, data enumerators approached the next household on the list to select an eligible participant.

## Data collection tool

A semi-structured questionnaire was developed as the data collection instrument. The questionnaire had three sections: particulars of the participants, socio-demographic characteristics, and 25 statements to evaluate the perception regarding child marriage. The statements were selected based on literature reviews [6,26,38,45,46] and expert opinions. We later organized these statements into four domains- social (8 statements), economic (4 statements), psychological (6 statements), and environmental (7 statements) covering the concept, causes, and consequences of child marriage in the context of rural Bangladesh during data analysis by reviewing literature and using thematic analysis [45,47].

The social domain included the societal, cultural, and community-related factors governing the perception regarding child marriage. These factors include sociocultural norms, social roles and expectations (i.e., based on age, gender, and social position), the role of social institutions (i.e., family, school), and community beliefs and practices. The economic domain included financial aspects, stimulating the perception regarding child marriage. It incorporated possible financial gain, loss, liability, and burden of the households occurring about or as a consequence of child marriage. The psychological domain covered mental and emotional aspects, comprising the factors, including expressing the internal beliefs, feelings, attitudes, deep understanding, and awareness of participants about the concept and consequences of child marriage. Finally, factors related to the living environment of the participants influencing child marriage-related perception were considered in the environmental domain. Those factors include security in the family and social environment, peer pressure, and legal issues. We obtained face validation of the instrument using expert opinions [48]. We pretested the instrument in another rural setting among 100 participants (50 parents and 50 adolescents). Based on the findings of pretesting, we made the necessary modifications to finalize the research instrument.

Item total correlation and domain-by-domain correlation tests found Pearson's 'r' value ranging from 0.06 to 0.61 and 0.60 to 0.72 (S1 and S2 Tables), and all were significant ($p < 0.05$). We tested the Cronbach's alpha to measure the internal consistency, and the obtained value was 0.68, which is considered adequate for conducting human-related research [49].

## Data collection

We started data collection on 1st April and completed it by 30th April 2023. The study used a pretested semi-structured questionnaire translated into the local Bangla language to conduct interviews. At first, data enumerators went to the center of the central market of selected unions, from where they took the roads going in different directions to cover the whole union. Two data enumerators formed a team to approach each selected household located in their chosen path, one to interview the adolescent and the other to interview either parent (father/mother) of that adolescent. In the same household, both interviews were conducted separately from each other to avoid any bias. Each face-to-face interview lasted for about 20–30 minutes.

## Data processing and statistical analysis

Data enumerators filled out the questionnaire and checked carefully for completeness and consistency of data immediately after each interview. After collection, data were coded, re-coded, and then entered into the computer for analysis using SPSS software (26th version). We calculated a wealth index to assess the socioeconomic status of the households, reflecting their

cumulative living standard. The index was calculated based on information regarding household durable goods and assets-refrigerator, mobile phone, television, bicycle, motor vehicle, furniture (bed, chair, table); housing characteristics- housing condition (based on the construction material of floor, roof and wall), and person per bedroom; access to utility services- source of drinking water, toilet facility, electricity, internet connection, and cable connection; and livestock, poultry assets. At first, we excluded the items that showed no significant variation using frequency analysis. We assigned a weighted score for each of the remaining items using principal component analysis. The total scores were divided into five equal parts or wealth quintiles. The 1st and last quintiles indicated the poorest and the richest households, respectively [50].

The socio-demographic characteristics and perception categories were presented in frequency and percentage. We explicitly checked the consistency and relevancy of data and took necessary measures like verifying, revisiting the participant, and re-interviewing through their contact cell phone number as required. We performed a normality test of the quantitative data and presented it by frequency, percentage, mean(±SD), and median (IQR) as applicable. The statistical significance of the difference in perception regarding child marriage was tested using the chi-square test. For categorizing the perception, we organized responses to 25 perception-related statements into a 2-level Likert scale, "Agree" and "Disagree". Values of 1 and 0 were assigned for each favorable and unfavorable response. The total score of perception ranged between 0 and 25 As per Bloom's cut-off points, participants with a score ≥20 (≥80%) indicated not supporting child marriage, whereas a lower score indicated support for child marriage [51]. Socio-demographic factors were significantly associated with perception regarding child marriage in binary analysis. The multi-nominal logistic regression model found the strength of the association of the factors with average and poor perception. Good perception was considered as the reference category. The significance level was 5%, and a p-value <0.05 was considered significant.

### Ethical considerations

We obtained ethical approval from the Institutional Review Board (IRB) of the National Institute of Preventive and Social Medicine (Reference No: NIPSOM/IRB/2023/09 Date: 12.01.2023). Before data collection, we got permission from the Chairman of the respective Union Parishad. We described the purpose and procedure of the study to all the participants in detail. Each parent participant signed an informed written consent form for their participation and to allow their adolescents to participate. In addition, each adolescent participant signed an informed written assent form. We ensured confidentiality, anonymity, voluntary participation, harmlessness, and the right to withdraw consent of all participants. We used data only for this study, with restricted accessibility to the principal investigator.

### Results

We attended 1087 households of selected unions and identified 1054 adolescents and parents as eligible participants. After that, we invited 1054 (527 adolescents and 527 parents) to participate in the study. Considering the unwillingness to participate (12, 1.1%), unavailability (8, 0.8%), and incomplete interviews (4, 0.4%), 1030 participants (515 adolescents and 515 parents) were enrolled in the study. Among the participants, 93.0% were Muslims and 7.0% were Hindus. The majority (81.6%) of the participants belonged to nuclear families. Around half (49.3%) had a monthly family income between 5000 and 20000 BDT, while 5.4% had a monthly family income of more than 50000 BDT. The median(IQR) monthly family income of the participants was 22,000 (15,000–30,000) BDT. Regarding the socio-economic status based on the wealth index, 19.8% of participants belonged to the poorest and 22.3% belonged to the richest group [Table 1]. Adolescents were 12–19 years old, and more than half (51.0%) of the parents were in the age group 20–39 years. The median (IQR) age of adolescents was 15 [23,17] years, while the median (IQR) age of parents was 38 [35–45] years. Around two-thirds (62.5%) of adolescents and three-fourths (75.3%) of parents were female. Regarding educational qualification, 67.5% of adolescents and 56.1% of parents had a secondary level of education. In respect of occupation, the majority (97.7%) of adolescents were students, while a maximum (63.5%) of parents were homemakers [Table 2].

**Table 1. Distribution of participants by common socio-economic characteristics (n = 1030).**

| Characteristics | Frequency | Percentage |
|---|---|---|
| **Religion** | | |
| Muslim | 958 | 93.0 |
| Hindu | 72 | 7.0 |
| **Family type** | | |
| Nuclear | 840 | 81.6 |
| Joint | 190 | 18.4 |
| **Monthly family income (BDT)** | | |
| 5000-20000 | 508 | 49.3 |
| 21000-49000 | 466 | 45.2 |
| 50000−100,000 | 56 | 5.4 |
| Median (IQR) | 22,000 (15,000-30,000) | |
| **Socio-economic status (Wealth Index in Quintiles)** | | |
| First | 204 | 19.8 |
| Second | 230 | 22.3 |
| Third | 156 | 15.1 |
| Fourth | 210 | 20.4 |
| Fifth | 230 | 22.3 |

BDT= Bangladeshi Taka, the currency of Bangladesh; IQR= Interquartile range.

**Table 2. Distribution of participants by background characteristics (n = 1030).**

| Background characteristics | Adolescents (n = 515) f(%) | Parents (n = 515) f(%) | Total (n = 1030) f(%) |
|---|---|---|---|
| **Age category (Years)** | | | |
| 12-19 | 515 (100.0) | – | 515 (50.0) |
| 20-39 | – | 263 (51.0) | 263 (25.5) |
| 40-69 | – | 252 (49.0) | 252 (24.5) |
| Median (IQR) | 15 (23–17) | 38 (36–4435–45) | |
| **Gender** | | | |
| Male | 193 (37.5) | 127 (24.7) | 320 (31.1) |
| Female | 322 (62.5) | 388 (75.3) | 710 (68.9) |
| **Level of education** | | | |
| No formal schooling | 1 (0.2) | 34 (6.6) | 35 (3.4) |
| Primary level | 109 (21.2) | 150 (29.1) | 258 (25.1) |
| Secondary level | 349 (67.8) | 289 (56.1) | 638 (61.9) |
| Higher secondary level and above | 56 (10.9) | 42 (8.2) | 98 (9.5) |
| **Occupation** | | | |
| Service | 3 (0.6) | 67 (13.0) | 70 (6.8) |
| Student | 503 (97.7) | 0 (0.0) | 503 (48.8) |
| Business | 0 (0.0) | 80 (15.5) | 80 (7.8) |
| Homemaker | 0 (0.0) | 327 (63.5) | 327 (31.7) |
| Farming and day labor | 9 (1.7) | 41 (8.0) | 50 (4.9) |
| **Marital status** | | | |
| In marital relationship | 4 (0.8) | 502 (97.5) | 506 (49.1) |
| Not in marital relationship | 511 (99.2) | 13 (2.5) | 524 (50.9) |

f= Frequency; %= Percentage.

Regarding domain-specific perception, adolescents and parents significantly (p<0.05) differed in terms of child marriage occurring due to parents' decisions in the social domain. On the other hand, in the economic domain, 83.1% of adolescents and 89.5% of parents mentioned that child marriage brings poor consequences to the family, and this difference in perception between both groups was statistically significant (p<0.05). In the psychological domain, 92.6% of adolescents and 96.3% of parents perceived child marriage as a burning issue, and this difference in perception was statistically significant (p<0.05). Concerning poor health consequences, 94.6% of adolescents and 97.6% of parents claimed that it can occur due to child marriage, and this difference in perception was also statistically significant (p<0.05). In the environmental domain, significant (p<0.05) differences in perception for all statements were observed between adolescents and parents [Table 3].

The study found that 26.4% of the participants supported child marriage. Regarding the participant type, support for child marriage was found in 25.8% and 27.0% of adolescents and parents, respectively [Fig 2].

The study found that 40% of the participants with no formal schooling supported child marriage in comparison to 17.3% with higher secondary and above level education. The level of education was significantly (p<0.05) associated with perception regarding child marriage. Compared to 25% of participants belonging to nuclear families, 32.6% of participants

**Table 3. Comparison of domains and attributes of perception regarding child marriage between adolescents and parents (n=1030).**

| Domains | Attributes | Adolescents (n=515) Agree f (%) | Parents (n=515) Agree f (%) | Significance |
|---|---|---|---|---|
| Social | CM occurs due to social norm | 244 (47.4) | 238 (46.2) | 0.70 |
| | Female vulnerability | 500 (97.1) | 495 (96.1) | 0.39 |
| | Causes interruption of education | 488 (94.8) | 496 (96.3) | 0.22 |
| | Have poor social impact | 441 (94.6) | 448 (87.0) | 0.52 |
| | Occurs due to parents' decision | 446 (86.6) | 418 (81.2) | **0.01** |
| | Reduced by adolescents' involvement | 383 (74.4) | 381 (74.0) | 0.88 |
| | Preventable by women empowerment | 352 (68.3) | 348 (67.6) | 0.78 |
| | Permitted in my religion | 83 (16.1) | 86 (16.7) | 0.80 |
| Economic | Brings poor economic consequences | 428 (83.1) | 461 (89.5) | **0.003** |
| | CM imposes poverty | 416 (80.8) | 416 (80.8) | 1.00 |
| | Economically beneficial to the family | 54 (10.5) | 45 (8.7) | 0.34 |
| | Helps to escape dowry | 225 (43.7) | 247 (48.0) | 0.16 |
| Psychological | Perceives legal age at marriage | 424 (82.3) | 436 (84.7) | 0.31 |
| | Perceives as a burning issue | 477 (92.6) | 496 (96.3) | **0.01** |
| | Favors CM | 8 (1.6) | 10 (1.9) | 0.63 |
| | CM is more acceptable than late marriage | 47 (9.1) | 49 (9.5) | 0.83 |
| | CM increases the possibility to get better spouse | 69 (13.4) | 112 (21.7) | **<0.001** |
| | CM related to poor health consequences | 487 (94.6) | 504 (97.9) | **<0.001** |
| Environmental | Causes conjugal maladjustment | 447 (86.8) | 478 (92.8) | **0.001** |
| | Induces domestic violence | 442 (85.8) | 474 (92.0) | **0.001** |
| | CM is illegal | 485 (94.2) | 507 (98.4) | **<0.001** |
| | CM is a punishable offence | 475 (92.2) | 502 (97.5) | **<0.001** |
| | Ensures security in living environment | 113 (21.9) | 169 (32.8) | **<0.001** |
| | Prevents negative attitude of the society | 122 (23.7) | 156 (30.3) | **0.01** |
| | Prevents pre-marital relations | 176 (34.2) | 210 (40.8) | **0.02** |

f= Frequency; %= Percentage; p<0.0= Significant at 95% CI, CM= Child Marriage.

Total and domain-specific perception scores did not differ significantly (p>0.05) between adolescents and parents [Table 4].

**Table 4. Comparison of perception scores between adolescents and parents across domains of perception regarding child marriage (n = 1030).**

| Domains | Perception score | | Significance |
|---|---|---|---|
| | Adolescents Median (IQR) | Parents Median (IQR) | |
| Social | 7.0 (6.0-7.0) | 7.0 (6.0-7.0) | 0.29 |
| Economic | 3.0 (3.0-4.0) | 3.0 (3.0-4.0) | 0.63 |
| Psychological | 6.0 (5.0-6.0) | 6.0 (5.0-6.0) | 0.55 |
| Environmental | 6.0 (5.0-7.0) | 6.0 (5.0-7.0) | 0.14 |
| Total | 21.0 (19.0-23.0) | 21.0 (19.0-23.0) | 0.37 |

p value was obtained by Mann-Whitney U test.

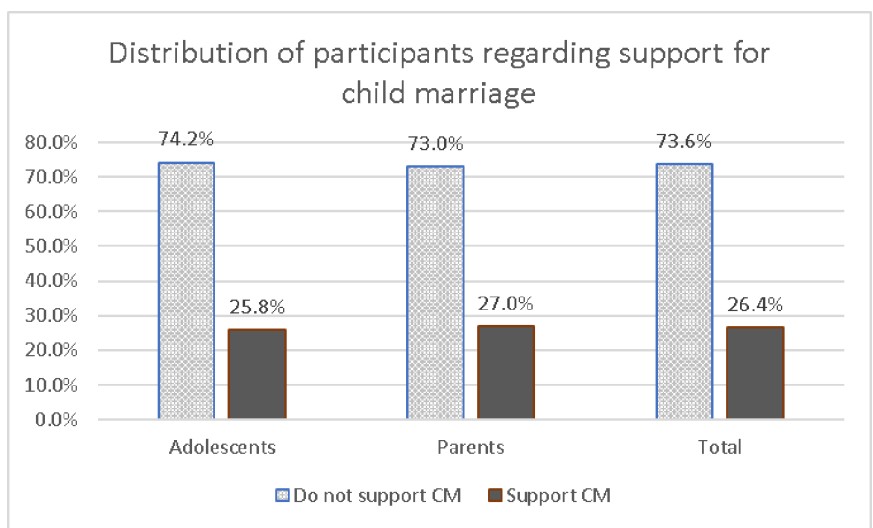

**Fig 2. Distribution of participants regarding support for child marriage.** The study compared the perception regarding child marriage between parents of adolescent boys and adolescent girls. However, no significant difference was found in any of the domains between those two groups of parents [Table 5].

from joint families supported child marriage, and the perception regarding child marriage was significantly (p < 0.05) associated with the family type of the participants [Table 6].

The study revealed that among the participants with no formal schooling and supporting child marriage, 92.9% were parents compared to 7.1% from the adolescent group. Support for child marriage significantly (p < 0.05) differed between the groups across levels of education. Among the child marriage supporter males, nearly two-thirds (63.8%) were adolescents, and among child marriage supporter females, 59.0% were parents. Both groups supporting child marriage differed significantly (p < 0.05) across gender [Table 7].

Participants with no formal schooling and primary level education were 3 (AOR-3.00, 95%; CI 1.27–7.09) times and 2.6 (AOR-2.69, 95%; CI 1.50–4.81) times more likely to support child marriage compared to participants with higher secondary and above level of education [Table 8].

## Discussion

Child marriage is one of the leading social problems in Bangladesh. Despite decreasing trends, child marriage is still remarkably prevalent in the country, especially in rural areas. The current study included rural adolescents and parents to

**Table 5.** Comparison of perception regarding child marriage between parents of adolescent boys and parents of adolescent girls (n = 515).

| Domains | Attributes | Parents of adolescent boys (n = 193) | Parents of adolescent girls (n = 322) | Significance |
|---------|-----------|------------------|------------------|-------------|
| | | Agree f (%) | Agree f (%) | |
| Social | CM occurs due to social norm | 85 (44.0) | 153 (47.5) | 0.44 |
| | Female vulnerability | 184 (95.3) | 311 (96.6) | 0.47 |
| | Causes interruption of education | 187 (96.9) | 309 (96.0) | 0.58 |
| | Have poor social impact | 164 (85.0) | 284 (88.2) | 0.29 |
| | Occurs due to parents' decision | 154 (79.8) | 264 (82.0) | 0.53 |
| | Reduced by adolescents' involvement | 143 (74.1) | 238 (73.9) | 0.88 |
| | Preventable by women empowerment | 128 (66.3) | 220 (68.3) | 0.63 |
| | Permitted in my religion | 31 (16.1) | 55 (17.1) | 0.80 |
| Economic | Brings poor economic consequences | 171 (88.6) | 290 (90.1) | 0.60 |
| | CM imposes poverty | 157 (81.3) | 259 (80.4) | 0.79 |
| | Economically beneficial to the family | 15 (7.8) | 30 (9.3) | 0.54 |
| | Helps to escape dowry | 89 (46.1) | 158 (49.1) | 0.51 |
| Psychological | Perceives legal age at marriage | 167 (86.5) | 269 (83.5) | 0.36 |
| | Perceives as a burning issue | 187 (96.9) | 309 (96.0) | 0.58 |
| | Favors CM | 4 (2.1) | 6 (1.9) | 1.00[a] |
| | CM is more acceptable than late marriage | 23 (11.9) | 26 (8.1) | 0.15 |
| | CM increases the possibility to get better spouse | 50 (25.9) | 62 (19.3) | 0.07 |
| | CM related to poor health consequences | 190 (98.4) | 314 (97.5) | 0.54[a] |
| Environmental | Causes conjugal maladjustment | 181 (93.3) | 297 (92.2) | 0.51 |
| | Induces domestic violence | 179 (92.7) | 295 (91.6) | 0.64 |
| | CM is illegal | 189 (97.9) | 318 (98.8) | 0.48[a] |
| | CM is a punishable offence | 187 (96.9) | 315 (97.8) | 0.56[a] |
| | Ensures security in living environment | 63 (32.6) | 106 (32.9) | 0.94 |
| | Prevents negative attitude of the society | 59 (30.6) | 97 (30.1) | 0.91 |
| | Prevents pre-marital relations | 83 (43.0) | 27 (39.4) | 0.42 |

CM= Child marriage, f= Frequency; %= Percentage; p<0.0= Significant at 95% CI; [a]p value for Fisher's Exact Test.

compare the level of perception regarding child marriage between two successive generations of rural Bangladesh. The study also intended to assess the perception regarding child marriage in respect of social, economic, psychological, and environmental dimensions.

## Perception of parents and adolescents regarding child marriage

The current study found supportive perception regarding child marriage in more than one-fourth of rural adolescents and their parents, which is very similar to the findings of another study conducted in Indonesia [26]. Both studies reported an insignificant difference between the two successive generations in terms of support for child marriage. Another study in Indonesia reported that 61.1% of parents and 47.6% of adolescents showed support for child marriage [52], which is much higher than our findings. Studies in Iran and Bangladesh reported much lower proportions of 5.1% and 11% of adolescents supporting child marriage [45,53]. The later studies only enrolled adolescent girls and used different tools to measure perception, which may contribute to the difference in findings of the present study. Regarding parents, a study in Ethiopia revealed that supportive perception was prevalent among half of them [54]. Regional and cultural differences may be attributable to these differences in findings.

Table 6. Comparison of perception regarding child marriage by socio-demographic features (n = 1030).

| Characteristics of the participants | Perception regarding child marriage | | | Significance |
|---|---|---|---|---|
| | Support Child Marriage f (%) | Do not support Child Marriage f (%) | Total f (%) | |
| **Type of participants** | | | | |
| Adolescents | 133 (25.8) | 382 (74.2) | 515 (100.0) | $\chi^2$ = 0.18 |
| Parents | 139 (27.0) | 376 (73.0) | 515 (100.0) | p = 0.67 |
| **Gender** | | | | |
| Male | 94 (29.4) | 226 (70.6) | 320 (100.0) | $\chi^2$ = 5.01 |
| Female | 178 (25.1) | 532 (74.9) | 710 (100.0) | P = 0.08 |
| **Level of education** | | | | |
| No formal schooling | 14 (40.0) | 21 (60.0) | 35 (100.0) | $\chi^2$ = 24.21 |
| Primary level | 94 (36.3) | 165 (63.7) | 259 (100.0) | **P = < 0.001** |
| Secondary level | 147 (23.0) | 491 (77.0) | 638 (100.0) | |
| Higher secondary level and above | 17 (17.3) | 81 (82.7) | 98 (100.0) | |
| **Occupation** | | | | |
| Service | 13 (13.6) | 57 (81.4) | 70 (100.0) | $\chi^2$ = 5.59 |
| Study | 128 (25.4) | 375 (74.6) | 503 (100.0) | P = 0.23 |
| Business | 20 (25.0) | 60 (75.0) | 80 (100.0) | |
| Homemaker | 93 (28.4) | 234 (71.6) | 327 (100.0) | |
| Farming and day labor | 18 (36.0) | 32 (64.0) | 50 (100.0) | |
| **Marital status** | | | | |
| In marital relationship | 137 (27.1) | 369 (72.9) | 506 (100.0) | $\chi^2$ = 0.22 |
| Not in marital relationship | 135 (25.8) | 389 (75.5) | 524 (100.0) | P = 0.63 |
| **Religion** | | | | |
| Islam | 258 (26.9)) | 700 (73.1) | 958 (100.0) | $\chi^2$ = 1.93 |
| Sanatan | 14 (19.4) | 58 (80.6) | 72 (100.0) | P = 0.16 |
| **Type of family** | | | | |
| Nuclear | 210 (25.0) | 630 (75.0) | 840 (100.0) | $\chi^2$ = 4.64 |
| Joint | 62 (32.6) | 128 (67.4) | 190 (100.0) | **P = 0.03** |
| **Socio-economic status (Wealth Index in Quintiles)** | | | | |
| First | 63 (30.9) | 141 (69.1) | 204 (100.0) | $\chi^2$ = 5.77 |
| Second | 53 (23.0) | 177 (77.0) | 230 (100.0) | P = 0.21 |
| Third | 34 (21.8) | 122 (78.2) | 156 (100.0) | |
| Fourth | 56 (26.7) | 154 (73.3) | 210 (100.0) | |
| Fifth | 66 (28.7) | 164 (71.3) | 230 (100.0) | |

f= Frequency, %= Percentage, $\chi^2$ = Chi-square; p<0.05= Significant at 95% CI.

Among the males who supported child marriage, the proportion of adolescents was higher compared to the parents (fathers). On the other hand, female parents (mothers) constituted a larger portion of females supporting child marriage. Abdurahman D et al., 2022 reported that a higher proportion of male parents supported child marriage, which is conflicting with our finding [54]. Compared to the fathers, the mothers remained far more concerned about their daughters and family honor. Insecurity for their young daughters in terms of sexual harassment and premarital relationships makes mothers more anxious, which propels them towards early marriage of their daughters despite knowing its grave consequences. Moreover, mothers in rural Bangladesh have to tolerate more negative words and

**Table 7. Comparison between adolescents and parents supporting child marriage across gender, level of education, marital status and family type (n = 1030).**

| Attributes | Support Child Marriage | | | Sig |
|---|---|---|---|---|
| | Adolescents f(%) | Parents f(%) | Total f(%) | |
| **Level of education** | | | | |
| No formal schooling | 1 (7.1) | 13 (92.9) | 14 (100.0) | **0.006** |
| Primary level | 43 (43.7) | 51 (54.3) | 94 (100.0) | |
| Secondary level | 81 (55.1) | 66 (44.9) | 147 (100.0) | |
| Higher secondary level and above | 8 (47.1) | 9 (52.9) | 17 (100.0) | |
| **Marital status** | | | | |
| In marital relationship | 4 (2.9) | 133 (97.1) | 137 (100.0) | **<0.001** |
| Not in marital relationship | 129 (95.6) | 6 (4.4) | 135 (100.0) | |
| **Type of family** | | | | |
| Nuclear | 100 (47.6) | 110 (52.4) | 210 (100.0) | 0.43 |
| Joint | 33 (53.2) | 29 (46.8) | 62 (100.0) | |
| **Gender** | | | | |
| Male | 60 (63.8) | 34 (36.2) | 94 (100.0) | **<0.001** |
| Female | 73 (41.0) | 105 (59.0) | 178 (100.0) | |

f = Frequency, % = Percentage, Chi-square test used to measure p value, p < 0.05 = Significant at 95% CI level.

**Table 8. Determination of factors associated with support for child marriage (based on univariable and multivariable logistic regression, n = 1030).**

| Factors | Categories | COR | 95% CI | P-value | AOR | 95% CI | P-value |
|---|---|---|---|---|---|---|---|
| Level of education | Higher secondary and above | Reference | | | Reference | | |
| | No formal schooling | 3.17 | 1.35-7.46 | **0.008** | 3.00 | 1.27-7.09 | **0.01** |
| | Primary level | 2.71 | 1.51-4.85 | **0.001** | 2.69 | 1.50-4.81 | **0.001** |
| | Secondary level | 1.42 | 0.81-2.48 | 0.20 | 1.43 | 0.82-2.50 | 0.20 |
| Family type | Nuclear | Reference | | | Reference | | |
| | Joint | 1.45 | 1.03-2.04 | **0.03** | 1.36 | 0.96-1.92 | 0.08 |

COR= Crude Odd's Ratio, AOR= Adjusted Odd's Ratio, CI= Confidence Interval; p<0.05= Significant at 95% CI.

attitudes of the society for their daughters' delayed marriage and possible promiscuity [55]. However, more exposure to education and information technologies has created awareness regarding the grave consequences of child marriage, resulting in lower support among female adolescents [45]. Assuming different roles in the family and society, differences in life experience create opposite perceptions among females of two successive generations. In comparison to adolescent boys, girls struggle more and face difficulties as young brides. In addition, married girls, compared to boys, get fewer opportunities to continue their studies, leading to the development of more anti-child marriage perceptions.

Both participant groups showed statistically indistinguishable overall and domain-specific perception scores. However, considering the individual statement, their perception significantly varied in all those in the environmental domain and a few in other domains. Differences in outlook, knowledge, and life experiences between two successive generations on child marriage can be attributable to these differences. However, the study found no significant difference in any statement between parents of adolescent boys and girls, reflecting similar awareness and experience among them.

## Perception of adolescents and parents in respect of social dimension

The majority of the participants perceived that child marriage occurs due to social norms, and almost all agreed that females are more vulnerable to this. A very popular saying in Bangladesh about women is "kuritei buri," meaning women get older at the age of 20, reflecting a strong social norm as well as pressure in favor of child marriage for females, and delaying marriage drastically reduces their value as brides [56]. Marriage in developing countries is more of a social fact and ritual than an individual and family concern. In Bangladesh, social, religious, and cultural values are more emphasized than individual preference and happiness for marriage. Girls are more often subjected to marriage at a very early age without paying heed to their opinions [57]. Furthermore, girls' chastity is related to family honor and dignity. Restrictions on girls' out-of-home movement, limiting education, and giving early marriage are considered effective steps to protect their dignity. Such types of social beliefs and norms influence perception and decision-making in favor of child marriage, especially for girls [19].

Nearly all participants in both groups agreed that child marriage was associated with the interruption of education and poor social impacts. Child marriage not only affects the individual level but also has dreadful consequences on families and society as well [45]. Apart from physical and emotional disturbances, evidence suggests that early marriage is associated with social isolation, disharmony in relationships, and narrowed access to social benefits like schooling and earning opportunities [58–60]. After marriage, some young girls have to stop their education [61,62]. Child marriage can lower social status and create an identity crisis for young married couples, as the interference by older family members is increased and creates obstacles in family life and decision-making [45].

The study found a significant difference in perceptions regarding the role of parents' decisions in child marriage. The adolescents thought that parents impose their decision of marriage on their adolescent children, which they obey regardless of their own choices. The majority of the parents agreed to it, and some of them denied and blamed adolescents for their marriage in childhood, as they forced their parents or sometimes fled to marry their loved ones. A study in India showed that patriarchy, social norms, and imposing decisions on adolescents, especially on girls, ignoring their opinions, are the most common social factors favoring child marriage [19]. Age-old, sociocultural traditions and economic hardship sometimes force parents to decide on the early marriage of their girls. However, some parents who have knowledge and experience regarding poor health outcomes of child marriage decide against child marriage [57].

About three-fourths of adolescents and parents supported the fact that involving adolescents in the decision-making process of marriage would reduce the occurrence of child marriage. A recent study showed that involving adolescent girls in marital decision-making reduced the risk of child marriage by 34% [63]. Around two-thirds of adolescents and parents mentioned that empowering women would act as a protective factor against child marriage. Women's empowerment reduces gender inequality and enables a woman to take part in decision-making for her marriage, which can reduce the occurrence of child marriage [64]. Evidence suggests that working women are 55.0% less likely to be victims of child marriage in comparison to non-working and semi-skilled women [3].

About 1 in 6 adolescents, as well as parents, mentioned that child marriage is allowed in their religion. Religion is closely linked with faith and culture, which promote a family and social environment in which a girl or boy is born and brought up, and their perception regarding marital age evolves. No minimum age is set for marriage by the teachings of the religion, so some children become mentally ready for marriage at an early age [65]. A study in Bangladesh found Muslims have a 50.0% higher rate of child marriage than followers of other religions [3]. Social awareness programs addressing the sociocultural stigma as well as misconceptions favoring child marriage by engaging all sectors of the community can be effective intervention to improve perception regarding child marriage [56]. A randomized control trial in Bangladesh involving female adolescents showed that 18 month long awareness building training programs regarding gender role and rights, critical thinking, and decision making through group activities, discussion, videos, drama, role play and E-book reduced the risk of child marriage by 28% (AHR-0.72, 95% CI 0.59–0.88). This training also helped to improve the

communication skills of girls with guardians about sensitive issues like marriage, dowry, etc., which favored the expression of their desire about their life [66].

## Perception of adolescents and parents in respect of economic dimension

Significantly higher proportion of parents agreed that child marriage brings poor economic consequences to the family, reflecting a difference of viewpoint between two successive generations. Our findings were similar to a study that revealed that, according to parents, young girls are often married to unemployed young boys unprepared to take the responsibilities of their girls and grandchildren, leading to emotional, familial, and health-related problems [38]. Evidence suggests that marriage at a younger age is associated with poverty [67,68]. On the other hand, the economic status of the family is one of the crucial determinants of marriage at an early age for girls [30], expressing the reciprocal relationship between poverty and child marriage. A study showed that 13.8% of women married before 18 years are more likely to live below the poverty line in comparison to 10.1% of women married after 18 years [26].

More than half of adolescents and parents agreed that early marriage of girls helps to lower the amount or sometimes escape of dowry, which has to be paid by the bride's family. Dowry is deeply embedded and very commonly practiced in the Southeast Asian subcontinent. Assets in the form of dowry are offered to the groom's family as a part of the marriage. In this practice, the bride's family is interested in marrying their daughter at a younger age to reduce the amount of dowry. As the elder age of the bride is considered less suitable for marriage, it has to be compensated by paying a higher amount. On the other hand, the groom's family is interested in marrying their younger sons to a younger girl to get the assets early [46].

Although the support for the economic benefit of child marriage is very low, a slightly but insignificant higher proportion of adolescents agreed with this. Our findings were in line with an Indonesian study reporting that some adolescents viewed marriage as a way of improving the economic and social status of the family, which outweighs its harmful consequences. In prospect to the bride's family, unmarried girls over 18 years are a burden and marrying them off can resolve economic problems [26]. In rural Bangladesh, people with limited earning opportunities consider marrying young girls profitable due to lower dowry and fewer mouths to feed in the family, rather than delaying marriage with uncertain future outcomes and bearing higher educational costs [61]. Poor parents often prefer early marriage to protect their girls and other members of the family from food insecurity and social bullying [69].

Raising awareness about the poor economic consequences of child marriage is necessary, especially among adolescents, along with efforts to improve their as well as their families' economic condition. Skill based training and financial aid would serve the purpose by creating employment opportunities for adolescents [70]. A study in Bangladesh revealed that training on life skills like computer programs, photography, first aid, and entrepreneurship can reduce the 30% occurrence of child marriage among girls [66]. Moreover, earning women have more capacity to negotiate dowry [71].

## Perception between of adolescents and parents in respect of psychological dimension

Nearly 1 in 6 adolescents and 1 in 7 parents did not know the legal age at marriage. A study in Ethiopia showed that a lack of awareness regarding the legal age for marriage is associated with a 2.5 times higher risk of child marriage [63]. A small proportion of participants in both groups had a supportive mentality towards child marriage. These findings differ from another study conducted in India, which reported that 26.7% of women living in urban slums had a supportive attitude towards child marriage [62]. Differences in the study period and settings may be the reasons for this difference. Although the support for child marriage was very negligible, nearly one in ten adolescents and parents accepted it rather than late marriage. Overwhelming effects of social stigma towards elder unmarried women and their families, as well as economic constraints, force a portion of adolescents and parents to accept early marriage instead of delaying it for a long period. These findings show that knowledge is not enough for better perception and practice; intervention directed towards mitigating socio-cultural and economic drivers of child marriage is also essential.

There was significant disagreement between adolescents and parents regarding child marriage as a burning issue. More exposure to print and electronic media and a wider social network may play a crucial role in favor of parents' better perception than adolescents in this issue. Moreover, many families did not provide smartphones and internet connections to adolescents in fear of addiction to social media and poor academic performance, which may contribute to their lower awareness of these issues, demanding ample coverage of child marriage in academic and extra-academic activities at schools.

The study revealed that parents have a strong perception that child marriage helps to get better spouses, especially for girls. Younger girls are preferred for marriage as they are more likely to be virgins, and parents have much more knowledge and experience regarding this than adolescents. Adolescents tend to be less oriented to the negative health consequences of child marriage than parents. Early marriage often results in early and unplanned conception, which can be harmful for both the young mothers and their children [72], more often observed by parents compared to adolescents through the experience of their own or their surrounding ones. Basic scientific knowledge regarding female reproductive issues is necessary to understand such poor outcomes. Lack of awareness and shyness prevent adolescents from acquiring comprehensive knowledge on child marriage from their families, especially in rural areas. Schools, however, can be a good alternative and effective source; inadequate coverage of the curriculum of the primary and secondary levels of education in Bangladesh, as well as hesitancy to discuss this topic by teachers, can be held responsible for maintaining this knowledge gap among adolescents [73].

## Perception of adolescents and parents in respect of environmental dimension

This domain showed maximum disagreement between the two groups of participants, reflecting generational differences in perception developed through diverse practical life experiences. Compared to the adolescents, the parents were more oriented toward conjugal maladjustment and domestic violence caused as a result of child marriage. A study reported that 40% of women married before 18 years old suffered from family conflicts [62]. Another study reported that about 32% of women married before 18 years of age became victims of domestic violence [59]. A study in Kenya found that 36.0% of women married before 18 years justified beating by their husbands in comparison to 20% of women married after 18 years of age [74]. Another study in Bangladesh reported that marrying in adulthood significantly reduced the risk of intimate partner violence by 25% [37]. Adolescent brides have lower bargaining power in a household, which makes them more vulnerable to physical and psychological violence by intimate partners [75]. The parents, compared to adolescents, were also more concerned about the legal issues related to child marriage, as it is a violation of the law, which is subject to punishment. Lack of knowledge, awareness and practical experience among adolescents regarding the familial and legal consequences of child marriage may be attributable to our findings. It demands designing strategies and interventions suitable for adolescents using academic, electronic, and social media platforms to make them understand the familial and legal benefits of avoiding child marriage.

In contrast, adolescents had significantly favorable perceptions in terms of insecurity, negative social attitudes, and premarital relationships related to child marriage. Parents are more exposed to harsh comments as their girls grow older and remain unmarried, so they become concerned about saving the honor of their families. For this, they do not wait for the legal age of 18 years but rather give marriage to their children at a younger age. Moreover, social stigma frightens families that unmarried girls would remain such if not married off early [63]. However, a study in Indonesia reported that adolescents are more eager to accept child marriage in comparison to their parents. From a very early age, they know that, as a successor, they must protect the family honor and dignity, and older unmarried girls in any family are considered a sign of bad luck [26]. Such negative attitudes of society toward older girls create a sense of insecurity in terms of sexual harassment, premarital relationships, and conception.

In comparison to adolescents, a significantly higher proportion of parents perceived that child marriage ensures the social security of girls and reduces the chance of illegal premarital relations. In this regard, a study revealed that parents are more supportive of the marriage of their underage girls to escape shame in society in case of premarital

pregnancy [26]. A Nigerian study reported that nearly two-thirds of the girls were forced to marry prematurely by their parents to the boys who impregnated them [38]. However, some adolescents also prefer early marriage to protect themselves from sexual harassment, as repeated such incidents made them believe that next time, it would be their turn [26]. Community-based awareness programs should be designed to alleviate these negative social attitudes, facilitating child marriage by using social, cultural, and religious institutions [70]. Proper implementation of existing laws against child marriage, eve-teasing can create a secure environment for girls and their families. Reproductive and legal counselling to adolescents aimed at preventing premarital conception and related child marriage is crucial to improve perception [76].

### Factors associated with perception regarding child marriage

The current study revealed that at least the secondary-level education was necessary to develop perception against child marriage. Primary-level education did not alleviate the risk of supportive perception towards child marriage [42,58,77]. Another study revealed that fathers with no formal education and primary education are 2.3 and 3.3 times more likely to marry off their girls early, respectively, than their more educated peers [63]. The current study found that both groups of participants with primary-level education had significant variation in the perception of child marriage, reflecting practical knowledge and experience of the parents as one of the differentiating factors. Another study conducted in Bangladesh showed that women with higher education are 93.0% less likely to be married before 18 years. Men with higher education (52.0%) were less likely to support and practice child marriage in comparison to men with no formal education [3]. With each additional level of education beyond the primary level, the chance of child marriage for girls significantly reduces as they spend more time in academic settings [78]. Women with higher education have higher ambitions for jobs, for which they postpone their marriage before attaining economic stability and higher positions in the family, which increases their bargaining capacity to choose the time of their marriage [79].

Appropriate strategies must be taken to maximize enrollment and retention of students, especially girls, at the secondary and higher-secondary levels. The existing government incentives for girl students should be increased to an optimum level. A study in Bangladesh reported that the provision of tuition support to pass exams and develop communicative skills in English reduced the 25% risk of child marriage among adolescent girls [66]. Moreover, the health, familial, economic, and legal consequences of child marriage should be incorporated adequately in late primary and early secondary level academic curriculum as well as in adult education programs to improve perception regarding child marriage.

The current study depicted that the occupation of the participants was not significantly associated with the level of perception regarding child marriage. However, studies in Bangladesh and Nepal showed that the women's working status influences the age at first marriage. Women working in formal sectors generate new ideas and enough resources, which creates negative perceptions about child marriage [3,80]. We found that participants from joint families were more likely to support child marriage, however, it was not significant when adjusted for level of education. A recent study showed, living with relatives in joint or extended families increases the risk of child marriage by 1.5 times [63]. The lower number of members in nuclear families facilitates better educational and economic opportunities, which enable them to shape their perception regarding child marriage.

Different awareness-building programs and behavior-change communication tools can improve perception regarding child marriage, which in turn can reduce the occurrence of child marriage [81]. A recent study in Bangladesh found that edutainment through awareness-raising TV serials regarding child marriage assisted in developing favorable attitudes toward child marriage [82]. A community-level clustered randomized controlled trial in Bangladesh employed the community-based Tipping Point Intervention (TPI) among adolescents, addressing social norms favoring child marriage through cross-gender and generation dialogues and enhancing women's empowerment through increasing knowledge, skills, and capabilities to reduce child marriage. That study found that higher attendance in these sessions was associated with a significant reduction in the risk of child marriage [83].

The present study considered four selected rural unions, which are very close to the capital city, Dhaka, so the findings might not represent the entire rural areas of Bangladesh. Moreover, data regarding child marriage among the parents was not collected, which could have explored their own life experiences regarding child marriage. It could also be compared whether the change in perceptions was due to their own child marriage experience or not.

## Conclusion

Support for child marriage is still prevalent in rural areas of Bangladesh, demanding improvement in the perception level of both parents and adolescents. Individuals with higher education showed more support against child marriage in comparison to lower or not formally educated individuals. Although overall perception regarding child marriage did not differ between the two successive generations, significant disagreements were found in a few aspects, especially in the environmental domain, due to differences in life experience and outlook about this issue. Moreover, a higher proportion of male adolescents and female parents were found among the child marriage supporter group of participants, reflecting differences in role and interest within participants of the same gender across two generations. A higher proportion of adolescents had an unfavorable perception regarding child marriage as a burning issue, which is illegal and punishable, brings poor economic and health consequences, and is responsible for maladjustment and domestic violence. Adolescents were more concerned about the role of parents' decisions in favor of child marriage. A higher proportion of parents had unfavorable perceptions regarding child marriage, like protection from negative attitudes and premarital relationships, helping to ensure the security of girls, and getting a better spouse. Improving perception by education can be the first and foremost, but not the only step, for preventing child marriage. In addition to the academic curriculum, adolescents must be aware of the harmful health, economic, familial, social, and legal consequences of child marriage through suitable awareness programs. Inclusive social awareness programs, especially for less educated people, focusing on the benefits of avoiding early marriage, minimizing stigma and misconceptions using social and religious institutions, are essential to improve parents' perception. In addition, social and legal programs for mitigating the financial and social insecurity of girls can improve perception regarding child marriage.

## Supportive information

**S1 Table. Supplementary Table 1.**
(DOCX)

**S2 Table. Supplementary table 2.**
(DOCX)

**S3 Table. STROBE_Checklist_Cross-sectional_Child Marriage.**
(DOCX)

**S4 Table. Child marriage dataset.**
(XLSX)

## Acknowledgments

Authors thank all the participants for their sincere participation and cooperation during data collection.

## Author contributions

**Conceptualization:** Md. Ziaul Islam.

**Data curation:** Md. Ziaul Islam, S. M. Sharf-ul-alam, Farjana Farha, Nargis Sultana, Ananya Adhya, Sharmin Farjana.

**Formal analysis:** Md. Ziaul Islam, S. M. Sharf-ul-alam, Ananya Adhya.

**Investigation:** Md. Ziaul Islam, S. M. Sharf-ul-alam, Nargis Sultana, Sharmin Farjana.

**Methodology:** Md. Ziaul Islam, Ananya Adhya, Sharmin Farjana.

**Project administration:** Md. Ziaul Islam, Farjana Farha, Nargis Sultana.

**Resources:** Md. Ziaul Islam, Farjana Farha, Ananya Adhya, Sharmin Farjana.

**Software:** Md. Ziaul Islam, S. M. Sharf-ul-alam.

**Supervision:** Md. Ziaul Islam, Nargis Sultana, Ananya Adhya.

**Validation:** Md. Ziaul Islam, Sharmin Farjana.

**Visualization:** Md. Ziaul Islam, S. M. Sharf-ul-alam, Ananya Adhya.

**Writing – original draft:** Md. Ziaul Islam.

**Writing – review & editing:** Md. Ziaul Islam, S. M. Sharf-ul-alam, Farjana Farha, Nargis Sultana, Sharmin Farjana.

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
