## [Decision Letter · Decision Letter 0]

4 Nov 2024

Perception of rural adolescents and parents regarding child marriage: Findings of a community-based cross-sectional study in Bangladesh

PLOS ONE

Dear Dr. Islam,

Thank you for submitting your manuscript to PLOS ONE. After careful consideration, we feel that it has merit but does not fully meet PLOS ONE’s publication criteria as it currently stands. Therefore, we invite you to submit a revised version of the manuscript that addresses the points raised during the review process.

A rebuttal letter that responds to each point raised by the academic editor and reviewer(s). You should upload this letter as a separate file labeled 'Response to Reviewers'.A marked-up copy of your manuscript that highlights changes made to the original version. You should upload this as a separate file labeled 'Revised Manuscript with Track Changes'.An unmarked version of your revised paper without tracked changes. You should upload this as a separate file labeled 'Manuscript'.If applicable, we recommend that you deposit your laboratory protocols in protocols.io to enhance the reproducibility of your results. Protocols.io assigns your protocol its own identifier (DOI) so that it can be cited independently in the future. For instructions see: https://journals.plos.org/plosone/s/submission-guidelines#loc-laboratory-protocols . Additionally, PLOS ONE offers an option for publishing peer-reviewed Lab Protocol articles, which describe protocols hosted on protocols.io. Read more information on sharing protocols at https://plos.org/protocols?utm_medium=editorial-email&utm_source=authorletters&utm_campaign=protocols .

We look forward to receiving your revised manuscript.

Kind regards,

Nishith Prakash, Ph.D.

Academic Editor

PLOS ONE

Journal Requirements:

3. We note that you have indicated that there are restrictions to data sharing for this study. PLOS only allows data to be available upon request if there are legal or ethical restrictions on sharing data publicly. For more information on unacceptable data access restrictions, please see http://journals.plos.org/plosone/s/data-availability#loc-unacceptable-data-access-restrictions. Before we proceed with your manuscript, please address the following prompts: a) If there are ethical or legal restrictions on sharing a de-identified data set, please explain them in detail (e.g., data contain potentially identifying or sensitive patient information, data are owned by a third-party organization, etc.) and who has imposed them (e.g., a Research Ethics Committee or Institutional Review Board, etc.). Please also provide contact information for a data access committee, ethics committee, or other institutional body to which data requests may be sent. b) If there are no restrictions, please upload the minimal anonymized data set necessary to replicate your study findings to a stable, public repository and provide us with the relevant URLs, DOIs, or accession numbers. For a list of recommended repositories, please see https://journals.plos.org/plosone/s/recommended-repositories. You also have the option of uploading the data as Supporting Information files, but we would recommend depositing data directly to a data repository if possible. We will update your Data Availability statement on your behalf to reflect the information you provide.

Reviewers' comments:

Reviewer's Responses to Questions

**Comments to the Author**

1. Is the manuscript technically sound, and do the data support the conclusions?

Reviewer #1: No

Reviewer #2: Partly

2. Has the statistical analysis been performed appropriately and rigorously?

Reviewer #1: No

Reviewer #2: Yes

3. Have the authors made all data underlying the findings in their manuscript fully available?

Reviewer #1: No

Reviewer #2: Yes

4. Is the manuscript presented in an intelligible fashion and written in standard English?

Reviewer #1: Yes

Reviewer #2: No

Reviewer #1: 1. The authors investigate potential differences in attitudes towards child marriage between parents and adolescents who participated in the study. This is an interesting angle. However, the Background/Introduction section should provide a rationale for why examining these differences in perception and attitudes is important to explore.

2. A recent paper examines differences in attitudes towards child marriage in the context of an edutainment intervention. The authors may want to reference this article.

Islam, Muhammed Nazmul, Atonu Rabbani, Animesh Talukder, Rubaiya Riya Siddiqua, Sanjana Nujhat, Mushfiqur Rahman, Antara Roy, and Malabika Sarker. “Edutainment and the prevention of under-age marriages: The evaluation of a television series designed to promote positive role models in Bangladesh.” Journal of Development Effectiveness 16, no. 2 (2024): 159-186.

3. The study uses a survey questionnaire to understand perceptions of child marriage. It would be helpful to explain the basis for these questions. Did the authors adopt any pre-existing tools? While all domains seem reasonable, what is the theoretical justification for them? I recommend linking the questions to relevant literature to provide justification and a foundation for the survey instrument.

4. Related to #3, the authors should elaborate on what the different domains mean and how questions across different domains relate to the scope of each specific domain.

5. Related to #4 (and #3), how did the researchers validate the questions?

6. In Table 3, what do "good," "average," and "poor" perception mean? Do they indicate how favorable respondents feel about child marriage outcomes?

7. P.11, lines 257-262 seem to repeat information from the Methods section. I recommend incorporating this into the Methods section without repetition.

8. The rows in Table 4 add up to 100% within each type of respondent. Isn’t reporting either "yes" or "no" sufficient?

9. The authors have not established expectations for the quantitative findings, making it difficult to interpret Table 4. In most cases, especially for the first three "domains," there doesn't seem to be much difference between the two types of respondents. Is this expected or not?

10. The only discernible differences appear in the "Environmental" domain. Why is this the case?

11. Authors should report the correlation of answers across domains and questions. There should be tests for internal reliability of the survey tools.

12. The Discussion section primarily repeats the findings. This section should be used to suggest implications of the results and relate the findings to the broader literature. I request that the authors revise it accordingly.

Reviewer #2: In this study, the authors assess the perception of child marriage among adolescents and parents, focusing on differences in perceptions across the two groups. They rely on a primary survey of 1030 participants residing in four sub-district administrative units in rural Bangladesh. The cross-sectional survey involved face-to-face interviews using a semi-structured questionnaire that aimed to collect information on respondents’ demographic and socio-economic characteristics as well as their responses to 25 statements that measured their perception towards child marriage on four dimensions- social, economic, psychological and environmental. The authors conduct a multinomial logistic regression to determine the factors influencing perceptions towards child marriage. This was complemented by chi-square analysis to test if the differences in perceptions across each of the attributes considered are statistically significant. The authors present two key results. Firstly, they find that perception towards child marriage differs significantly across individuals with different level of education, current marital status and type of family (joint or nuclear) while socio-economic status, occupation, religion and gender do not explain differences in perception towards child marriage. Secondly, the authors find that adolescents and parent perceptions on social, economic and psychological attributes of child marriage do not differ significantly, whereas perceptions regarding environmental attributes of child marriage differ significantly across the two groups.

The study deals with a relevant social issue that persists in the context of the study- Bangladesh, as well as in other developing countries. It is important to understand how individuals perceive child marriage and to identify factors that drive support towards a practice that has been empirically established to have negative socio-economic outcomes for women, and the society at large. The authors adopt an appropriate method to this inquiry by using a semi-structured interview to gather data on this sensitive topic as it offers important insights into individual perceptions which are often difficult to interpret. Collecting sensitive data from a sample of 1030 individuals, along with useful demographic and socio-economic characteristics provides for a rich source of data. I also appreciate the safeguards the authors have adopted in carrying out this survey. By ensuring the surveys are conducted in anonymity and in the absence of other family members, they prevent some potential bias in responses.

However, the paper is not without its shortcomings. The analysis could be developed further to throw light onto the interactive effects of the various socio-economic characteristics and differences across adolescents and parental perceptions. The objective of the study and therefore, its implications, could be made more nuanced and clarified. The authors do not explain in enough detail how the attributes were chosen and classified into the four domains used in the analysis. Similarly, while there is some discussion on the factors that drive the observed perceptions, it is not clear where these conclusions are drawn from. I discuss these shortcomings and suggestions for improvement below. Taking these into consideration, I recommend that the paper be re-assessed for publication after the authors have made major revisions to the current manuscript.

Major Revisions:

1. Abstract/Introduction: The abstract describes the main objective of the study to be to “assess the perception of rural adolescents and parents regarding child marriage”. However, from the introduction and the discussion of the results, it is evident that the study focusses sufficiently on a comparison of differences in perceptions across adolescents and parents. The research question and objective need to be clarified to highlight a) that the study aims to compare differences in perceptions on child marriage across adolescents and parents; b) that the objectives include identifying factors/characteristics that influence perceptions on child marriage

2. Sample selection: I recommend adding a note on the representative of the unions of rural Bangladesh. There is some discussion towards the end (in the last session) but it would be appropriate to discuss the same earlier on.

3. The steps undertaken to survey the respondents has been discussed in great detail. However, certain methodological approaches need further explanation. Currently, there is no discussion on why the four domains- social, economic, psychological and environmental have been chosen. Moreover, the definition of each of these domains, especially with regards to child marriage needs to be included. More details on how the 25 questions/attributes were chosen and classified into the four domains would help the readers understand the concepts better. I am especially finding it difficult to distinguish between the attributes classified under social and environmental domains.

4. Page 10: Lines 235-244- I understand the responses to the statements were coded on a binary scale. However, attributing “correct answers” to 1 and “wrong answers” to 0 is not straightforward. Firstly, with the statements being subjective in nature, I do not think it is appropriate to classify them as correct and wrong answers. A more scientific approach would be to rank responses on a standardized scale where a higher value indicates support against child marriage. I would also recommend including the logistic regression specification in the manuscript.

5. Throughout the paper, the terms “good” and “bad” perceptions of child marriage have been used. It would be better to define these terms to ensure that readers understand that “good perception” as referred to in the paper implies support against child marriage.

6. In the discussion of the results, the authors refer to dowry, social norms etc. as reasons for the unanimous support in favor for child marriage prevalent in the sample. However, there is no explanation to how these conclusions were drawn. Are these conclusions drawn from evidence from open ended questions included in the survey? If so, please include details of the same. In case open-ended questions or any form of narrative analysis was undertaken, please include more details of how this was carried out, how surveyors recorded these responses and how the analysis was carried out.

7. I infer that the two main take-aways from this paper are that certain characteristics such as education, family structure and marital status can explain differences in perceptions on child marriage, and that perceptions on how child marriage affects environmental attributes differ across adolescents and parents. I would highlight these findings with further clarity in the conclusion. Moreover, I think it is possible to conduct further analyses to explore the intersection of these two aspects. For instance, it would be interesting to see if perceptions differ across adolescents and adults, given a level of education or gender.

8. It might be useful to include a discussion of potential confounders of perception of child marriage. For example, it is possible that surveyed parents who were themselves subjected to child marriage might harbor different perceptions than those who did not. Distinguishing/controlling for these factors would help identify factors that affect differential perceptions better. It might also be worth conducting further analyses after distinguishing between parents with sons and daughters. Would expect to see differences in perceptions between the two sets of parents but it would be interesting to see if the factors that drive perceptions differ across them.

9. The introduction and conclusion emphasize the importance of including the negative effects of child marriage in school curriculum to help change perceptions. Although this might have the desired effects, there is no evidence directly arising out of the analysis in this paper that suggests this. I recommend avoiding drawing such conclusions that are not directly linked to the evidence arising out of the study. Similarly, on page 17 (lines 337-340), the authors imply that social pressure is driving child marriages in Bangladesh. However, it is not clear if the questionnaire dealt with questions of social validation. I recommend the authors refer to the evidence supporting this statement or refrain from making them.

Minor Revisions:

1. More citations required:

a. Page 3, Line 72

b. Page 4, Line 94-95

c. Page 5, line 114

d. Page 6, Line 141

2. Page 4, line 98: Prevalence of child marriage has been adequately discussed. I would recommend balancing this discussion out with more evidence from the literature on the negative effects of child marriage on women’s outcomes and socio-economic outcomes, in general.

3. Page 5: The discussion of religion as a mediating factor for individual perceptions can be clarified further. The hypothesis seems to be that religion is an important mediating factor but the discussion needs to be refined to highlight how religion is expected to affect perceptions.

4. Page 6: Lines 133-143- can be moved to the introduction where the prevalence of child marriage in developing countries is discussed. Could also consider expanding on this discussion of the negative consequences of child marriage and highlight how this is motivates the current study.

5. Page 19: Line 369- “son-in-law” to be changed to “daughter-in-law”

**Do you want your identity to be public for this peer review?** For information about this choice, including consent withdrawal, please see our Privacy Policy

Reviewer #1: No

Reviewer #2: No

---

## [Author Response · Author response to Decision Letter 1]

20 Dec 2024

Journal Requirements:

• Responses: Thanks for the valuable instructions. All the items are included with the submission

Responses: Thanks for the vital direction. The manuscript meets all PLOS ONE's style requirements

Responses: Thanks for the valuable guidelines. It is stated in the Methods section

3. We note that you have indicated that there are restrictions to data sharing for this study. PLOS only allows data to be available upon request if there are legal or ethical restrictions on sharing data publicly.

Responses: Thanks for the valuable instruction. It is stated that data will be available from the corresponding author upon reasonable request

Responses to Comments of Reviewer # 1

Comment 1. The authors investigate potential differences in attitudes towards child marriage between parents and adolescents who participated in the study. This is an interesting angle. However, the Background/Introduction section should provide a rationale for why examining these differences in perception and attitudes is important to explore.

Response: Thanks for the crucial comment. The rationale for examining the difference between adolescents and parents in perception regarding child marriage is provided at background section on pages 8-9. It is important to identify the differences to formulated adolescent and parent specific interventions through understanding their viewpoints developed on the basis of their knowledge and experiences.

Comment 2. A recent paper examines differences in attitudes towards child marriage in the context of an edutainment intervention. The authors may want to reference this article.

Islam, Muhammed Nazmul, Atonu Rabbani, Animesh Talukder, Rubaiya Riya Siddiqua, Sanjana Nujhat, Mushfiqur Rahman, Antara Roy, and Malabika Sarker. “Edutainment and the prevention of under-age marriages: The evaluation of a television series designed to promote positive role models in Bangladesh.” Journal of Development Effectiveness 16, no. 2 (2024): 159-186.

Response: Thanks for the vital comment. This article is referred in the discussion section at page 31 to exemplify possible interventions for improving the perception of parents and adolescents regarding child marriage.

Comment 3. The study uses a survey questionnaire to understand perceptions of child marriage. It would be helpful to explain the basis for these questions. Did the authors adopt any pre-existing tools? While all domains seem reasonable, what is the theoretical justification for them? I recommend linking the questions to relevant literature to provide justification and a foundation for the survey instrument.

Response: Thanks for the valuable comment. The data collection tool which included 25 statements to determine child marriage related perception was developed by the authors by extensive literature review and obtaining experts’ opinion. During data analysis phase, these statements were grouped in to 4 domains by sing thematic analysis [Data collection tool segment of Methods section, page-11]

Comment 4. Related to #3, the authors should elaborate on what the different domains mean and how questions across different domains relate to the scope of each specific domain.

Response: Thanks for the crucial comment. The meaning of each domain was operationalized by the authors and defined in Data collection tool segment of Methods section, page-11. the relation of each statement within each domain was analyzed by the authors with the help of experts.

Comment 5. Related to #4 (and #3), how did the researchers validate the questions?

Response: Thanks for the precious comment. Face validation of the tools was conducted by expert opinion. As we are not opting to build up a scale, this face validation could be enough as a validation process [Data collection tool segment of Methods section, pages 11-12].

Comment 6. In Table 3, what do "good," "average," and "poor" perception mean? Do they indicate how favorable respondents feel about child marriage outcomes?

Response: Thanks for the appreciated comment. Higher scores meaning good perception indication stronger support against child marriage, average perception means not so strong support against child marriage and poor perception means support for child marriage [Data processing and analysis section of methods, page 13].

Comment 7. P.11, lines 257-262 seem to repeat information from the Methods section. I recommend incorporating this into the Methods section without repetition.

Response: Thanks for the valuable comment. It is corrected accordingly.

Comment 8. The rows in Table 4 add up to 100% within each type of respondent. Isn’t reporting either "yes" or "no" sufficient?

Response: Thanks for the crucial comment. It could be done, the responses of ‘yes’ and ‘no’ are replaced by ‘agree’ and ‘disagree’ respectively. However, the same response like ‘Agree’ does not carry a specific value of 0 or 1 in all statements. To visualize the differences with more clarity, both agree and disagree frequency and percentage might be helpful.

Comment 9. The authors have not established expectations for the quantitative findings, making it difficult to interpret Table 4. In most cases, especially for the first three "domains," there doesn't seem to be much difference between the two types of respondents. Is this expected or not?

Response: Thanks for the vital comment. It is expected and can be explained by the same sources of knowledge regarding child marriage like schools and social media etc.

Comment 10. The only discernible differences appear in the "Environmental" domain. Why is this the case?

Response: Thanks for the valuable comment. This could be bi-chance or may be due to difference in live experience of family and society as parents have more experience than adolescents which is discussed in discussion section page-24.

Comment 11. Authors should report the correlation of answers across domains and questions. There should be tests for internal reliability of the survey tools.

Response: Thanks for the cherished comment. Correlation of answers across domains and questions were reported in supplementary tables 2 and 1 respectively showing significant correlation. Internal consistency was tested by Cronbach’s alpha test with obtained value of 0.68 [Data collection tool segment of Methods section, page-12].

Comment 12. The Discussion section primarily repeats the findings. This section should be used to suggest implications of the results and relate the findings to the broader literature. I request that the authors revise it accordingly.

Response: Thanks for the esteemed comment. The discussion section was revised accordingly.

Responses to Comments of Reviewer # 2

Comments: In this study, the authors assess the perception of child marriage among adolescents and parents, focusing on differences in perceptions across the two groups. They rely on a primary survey of 1030 participants residing in four sub-district administrative units in rural Bangladesh. The cross-sectional survey involved face-to-face interviews using a semi-structured questionnaire that aimed to collect information on respondents’ demographic and socio-economic characteristics as well as their responses to 25 statements that measured their perception towards child marriage on four dimensions- social, economic, psychological and environmental. The authors conduct a multinomial logistic regression to determine the factors influencing perceptions towards child marriage. This was complemented by chi-square analysis to test if the differences in perceptions across each of the attributes considered are statistically significant. The authors present two key results. Firstly, they find that perception towards child marriage differs significantly across individuals with different level of education, current marital status and type of family (joint or nuclear) while socio-economic status, occupation, religion and gender do not explain differences in perception towards child marriage. Secondly, the authors find that adolescents and parent perceptions on social, economic and psychological attributes of child marriage do not differ significantly, whereas perceptions regarding environmental attributes of child marriage differ significantly across the two groups.

The study deals with a relevant social issue that persists in the context of the study- Bangladesh, as well as in other developing countries. It is important to understand how individuals perceive child marriage and to identify factors that drive support towards a practice that has been empirically established to have negative socio-economic outcomes for women, and the society at large. The authors adopt an appropriate method to this inquiry by using a semi-structured interview to gather data on this sensitive topic as it offers important insights into individual perceptions which are often difficult to interpret. Collecting sensitive data from a sample of 1030 individuals, along with useful demographic and socio-economic characteristics provides for a rich source of data. I also appreciate the safeguards the authors have adopted in carrying out this survey. By ensuring the surveys are conducted in anonymity and in the absence of other family members, they prevent some potential bias in responses.

However, the paper is not without its shortcomings. The analysis could be developed further to throw light onto the interactive effects of the various socio-economic characteristics and differences across adolescents and parental perceptions. The objective of the study and therefore, its implications, could be made more nuanced and clarified. The authors do not explain in enough detail how the attributes were chosen and classified into the four domains used in the analysis. Similarly, while there is some discussion on the factors that drive the observed perceptions, it is not clear where these conclusions are drawn from. I discuss these shortcomings and suggestions for improvement below. Taking these into consideration, I recommend that the paper be re-assessed for publication after the authors have made major revisions to the current manuscript.

Major Revisions:

Comment 1. Abstract/Introduction: The abstract describes the main objective of the study to be to “assess the perception of rural adolescents and parents regarding child marriage”. However, from the introduction and the discussion of the results, it is evident that the study focusses sufficiently on a comparison of differences in perceptions across adolescents and parents. The research question and objective need to be clarified to highlight a) that the study aims to compare differences in perceptions on child marriage across adolescents and parents; b) that the objectives include identifying factors/characteristics that influence perceptions on child marriage.

Response: Thanks for the appreciated comment. Comparison in difference of perception between adolescents and parents and factors influencing perception regarding child marriage are added as objectives in both abstract [page-2] and in background [page 8&9].

Comment 2. Sample selection: I recommend adding a note on the representative of the unions of rural Bangladesh. There is some discussion towards the end (in the last session) but it would be appropriate to discuss the same earlier on.

Response: Thanks for the vital observation. A note on the representative of the unions of rural Bangladesh is added in sample size and sampling part of the methods on page 10.

Comment 3. The steps undertaken to survey the respondents has been discussed in great detail. However, certain methodological approaches need further explanation. Currently, there is no discussion on why the four domains- social, economic, psychological and environmental have been chosen. Moreover, the definition of each of these domains, especially with regards to child marriage needs to be included. More details on how the 25 questions/attributes were chosen and classified into the four domains would help the readers understand the concepts better. I am especially finding it difficult to distinguish between the attributes classified under social and environmental domains.

Response: Thanks for the valuable comment. Twenty-five statements were chosen by expert opinion and extensive literature review. The during data analysis, these statements were grouped in to four themes by thematic analysis by the researchers. The domains were defined and mentioned at ‘data collection section’ page-11.

Comment 4. Page 10: Lines 235-244- I understand the responses to the statements were coded on a binary scale. However, attributing “correct answers” to 1 and “wrong answers” to 0 is not straightforward. Firstly, with the statements being subjective in nature, I do not think it is appropriate to classify them as correct and wrong answers. A more scientific approach would be to rank responses on a standardized scale where a higher value indicates support against child marriage. I would also recommend including the logistic regression specification in the manuscript.

Response: Thanks for the valuable observation. Considering the subjectivity of answers, ‘yes’ and ‘no’ responses were replaced by ‘agree’ and ‘disagree’. Favourable response indicating support against child marriage was given 1 and unfavourable response was given 0 value. Higher value indicated strong support against child marriage [Data processing and statistical analysis section page, 13]. Logistic regression specification was added in page 13.

Comment 5. Throughout the paper, the terms “good” and “bad” perceptions of child marriage have been used. It would be better to define these terms to ensure that readers understand that “good perception” as referred to in the paper implies support against child marriage.

Response: Thanks for the vital observation. Corrected according to that [Data processing and statistical analysis section page- 13]

Comment 6. In the discussion of the results, the authors refer to dowry, social norms etc. as reasons for the unanimous support in favor for child marriage prevalent in the sample. However, there is no explanation to how these conclusions were drawn. Are these conclusions drawn from evidence from open ended questions included in the survey? If so, please include details of the same. In case open-ended questions or any form of narrative analysis was undertaken, please include more details of how this was carried out, how surveyors recorded these responses and how the analysis was carried out.

Response: Thanks for the vital comment. The difference of perception attributed by education, family type and marital status was mentioned in conclusion section. The difference of perception variance was more prominent in environmental domain also added in conclusion and the difference was likely due to difference of live experience between both groups [Page-32].

Comment 7. I infer that the two main take-away from this paper are that certain characteristics such as education, family structure and marital status can e

---

## [Decision Letter · Decision Letter 1]

8 Apr 2025

Thank you for your revised submission of the manuscript entitled “Perception of rural adolescents and parents regarding child marriage: Findings of a community-based cross-sectional study in Bangladesh.” We appreciate the thoughtful responses you provided to the initial round of reviewer comments, and we acknowledge the improvements made—particularly in the discussion section. However, both referees thought you need to do a better job and have highlighted their concerns. 

After a careful evaluation of the revised manuscript and the accompanying reviewer comments, we believe that additional revisions are necessary before the manuscript can be considered for publication. The reviewer has raised several substantive and methodological issues that should be addressed to strengthen the manuscript.

<h3 data-end="878" data-start="844">Key points requiring revision:</h3>

**Terminology Clarity** : The use of terms such as “poor” or “good” perceptions remains ambiguous. The reviewer strongly recommends replacing these with clearer terms such as “support” or “do not support” child marriage, to avoid misinterpretation.**Conceptual Framing of Statements** : The classification of survey items into domains needs further clarification. It is important to distinguish between statements about causes, consequences, and value judgments. Some statements currently included may reflect observations rather than normative views.**Sampling and Context** : Greater detail is needed on how the four unions were selected. Additionally, while we understand the need to protect participant identity, the manuscript would benefit from indicating the broad geographic location (e.g., district or upazila) if permissible under ethical guidelines.**Wealth Index Categorization** : The authors should clarify how wealth indices were constructed. If quintiles were used, it would be clearer to refer to them as such instead of using less precise terms like “middle class.”**Statistical Presentation** : Table 4 should be revised to present, for each characteristic, the percentage of parents and adolescents who support child marriage, along with corresponding p-values. This would improve the interpretability and relevance of the findings.**Discussion Section** : The discussion currently includes several statistical reiterations that belong in the results section. The authors should streamline this section and focus on interpretation and implications. The reviewer also encourages more detailed consideration of intervention strategies tailored to parents and adolescents, as well as reflection on why adolescents may exhibit more favorable attitudes toward child marriage than parents.**Additional Points** : Minor issues such as missing publication years in references should also be corrected.

In summary, while the manuscript shows promise and addresses an important issue, further revisions are necessary to enhance clarity, methodological transparency, and analytical depth. We encourage you to address each of the reviewer’s points thoroughly in your next revision. Since the reviewers have come up with "major revision" as a recommendation, I will appreciate if you address these concerns carefully in this round.

We look forward to receiving a revised version that incorporates these changes.

We look forward to receiving your revised manuscript.

Kind regards,

Nishith Prakash, Ph.D.

Academic Editor

PLOS ONE

Reviewers' comments:

Reviewer's Responses to Questions

**Comments to the Author**

Reviewer #1: (No Response)

Reviewer #2: (No Response)

2. Is the manuscript technically sound, and do the data support the conclusions?

Reviewer #1: Partly

Reviewer #2: Partly

3. Has the statistical analysis been performed appropriately and rigorously?

Reviewer #1: Yes

Reviewer #2: Yes

4. Have the authors made all data underlying the findings in their manuscript fully available?

Reviewer #1: No

Reviewer #2: Yes

5. Is the manuscript presented in an intelligible fashion and written in standard English?

Reviewer #1: Yes

Reviewer #2: No

Reviewer #1: Comments on the revised version of the manuscript entitled “Perception of rural adolescents and parents regarding child marriage: Findings of a community-based cross-sectional study in Bangladesh”

The authors have provided itemized responses to the comments received in the first round. I have several more suggestions that I believe will help the authors to improve the manuscript.

1. I apologize but I still find the terms “poor” perception towards child marriage ambiguous. Why not use the terms such as “Least Favorable” views towards child marriage which is much easy interpret?

2. This is related to a comment I had earlier about the justifying the relevance of the perceptions towards early child marriage. Aren’t there prior works that show how perceptions towards a social phenomenon such as early marriage determine actual outcomes?

3. Page 10, lines 240-1: How did the researchers choose four unions to carry out the study? Is the present research based on another existing or ongoing research?

4. Authors of the study also did not reveal the location of the study. Was it restricted by the IRB? Obviously, we don’t want the identities of the individual participants. But sharing the broad locations (district or upazila) can help the readers to contextualize the findings. Authors should consider this.

5. Page 14, lines 340-3: How were the wealth indexes categorized? Were the divided into quintiles? If so, the authors should just use quintiles instead of using less well-defined terms such as “middle class.”

6. Do the perceptions of the parents towards early child marriage vary by the sex of their children?

7. The authors repeat and mention lots of statistics in the discussion section. Many of them are redundant. They should be in the Findings section and authors should am to interpret the results more succinctly in the Discussion section. This will also help shortening the Discussion section.

8. [Minor comment] Page 7, lines 171-8: The authors referenced to a paper without the publication year. Authors may want to revise the sentence.

Reviewer #2: Thank you for addressing comments raised in the review. I appreciate the changes made to the manuscript, especially to the discussion section. Here are some more comments to help improve the paper:

1. The statement selection into the domains is unconvincing. Although there is more clarity on what the different domains are in the revised version, I think it is important to distinguish between statements that refer to causes and consequences of child marriage. Moreover, some statements such as "child marriage is a burning issue", "occurs due to parents" etc. simply indicate what respondents think is happening and agreeing to these statements may not necessarily mean they do not support child marriage.

2. Although you mention what "good" and "bad" perceptions are as used in the manuscript, I strongly advice you to change the terminology throughout the manuscript to "not support" and "support" child marriage. A good perception regarding child marriage could be mistaken to mean they perceive child marriage to be good and this is confusing for the reader.

3. Thank you for adding Table 4 in response to the previous comments on intersectionality. In its current form, Table 4 is telling us what percent of those who support (or not) child marriage are adults and adolescents for each characteristic. Why arent the p-values from the chi-square test not reported for each group then? It might be more insightful if you can rework this table to show for each given characteristic, what percent of parents and adolescents supported child marriage and if this difference is significant. For instance, among those with secondary education, how many adolescents and parents perceived child marriage an issue (rather than looking at what percent of those with secondary education and reported "good perception" were adolescents and parents).

4. In the discussion section, authors could include more implications from the study. Since a large part of the paper looks into differences in perceptions among 2 groups, who should the proposed awareness interventions target? Are there specific topics that need to be dealt with differently for either groups? How do the authors explain why the adolescents harbor more favorable stance in support of child marriage than parents?

**Do you want your identity to be public for this peer review?** For information about this choice, including consent withdrawal, please see our Privacy Policy

Reviewer #1: No

Reviewer #2: No

---

## [Author Response · Author response to Decision Letter 2]

10 May 2025

1. Terminology Clarity: The use of terms such as “poor” or “good” perceptions remain ambiguous. The reviewer strongly recommends replacing these with clearer terms such as “support” or “do not support” child marriage, to avoid misinterpretation.

Response: Thank a lot for the valued comment. Three categories of perception level; ‘good, ‘average’ and ‘poor’ have been replaced by two categories as “support” and “do not support” following your suggestion.

2. Conceptual Framing of Statements: The classification of survey items into domains needs further clarification. It is important to distinguish between statements about causes, consequences, and value judgments. Some statements currently included may reflect observations rather than normative views.

Response: Thanks for the pertinent comment. The concept behind four domains is described in details at Page-12, lines: 279-292. Some survey statements are rephrased to distinguish them clearly as cause, consequence or value judgement as applicable in Table-3 (pages-17-18) and Table-5 (pages-19-20).

At present, it is not possible to make all statements as normative views. Through extensive literature review, we found that most of the statements taken in this regards were observations and opinions of the participants rather normative views. Some relevant references of literature are given below-

Wibowo HR, Ratnaningsih M, Goodwin NJ, Ulum DF, Minnick E. One household, two worlds: Differences of perception towards child marriage among adolescent children and adults in Indonesia. The Lancet Regional Health–Western Pacific. 2021 Mar 1;8.

Naghizadeh S, Mirghafourvand M, Mohammadi A, Azizi M, Taghizadeh-Milani S, Ganbari H. Knowledge and viewpoint of adolescent girls regarding child marriage, its causes and consequences. BMC women's health. 2021 Dec;21:1-0.

3. Sampling and Context: Greater detail is needed on how the four unions were selected. Additionally, while we understand the need to protect participant identity, the manuscript would benefit from indicating the broad geographic location (e.g., district or upazila) if permissible under ethical guidelines.

Response: Thanks for the crucial comment. Four unions were selected as study places using multistage systematic random sampling technique, which is described in page 9 and in the page 10 under Figure -1.

4. Wealth Index Categorization: The authors should clarify how wealth indices were constructed. If quintiles were used, it would be clearer to refer to them as such instead of using less precise terms like “middle class.”

Response: Thanks a lot for the vital comment. The construction process is described with greater details in the page no 13. Wealth quintiles are used to categorize the socioeconomic class and mentioned as such according to your recommendation.

5. Statistical Presentation: Table 4 should be revised to present, for each characteristic, the percentage of parents and adolescents who support child marriage, along with corresponding p-values. This would improve the interpretability and relevance of the findings.

Response: Thanks for the valued comment. Table 4 (Pages, 18-19), and now Table 7 (Pages, 22-23), has been presented as per your recommendation.

6. Discussion Section: The discussion currently includes several statistical reiterations that belong in the results section. The authors should streamline this section and focus on interpretation and implications. The reviewer also encourages more detailed consideration of intervention strategies tailored to parents and adolescents, as well as reflection on why adolescents may exhibit more favorable attitudes toward child marriage than parents.

Response: Thanks a lot for the most crucial comment. In this current revised version, discussion section contains specific interpretation of the results with greater and detailed focus on implementation and intervention strategies for parents and adolescents

7. Additional Points: Minor issues such as missing publication years in references should also be corrected.

Response: Publication year is being corrected in reference list.

---

## [Decision Letter · Decision Letter 2]

26 Jun 2025

Dear Dr. Islam,

We look forward to receiving your revised manuscript.

Kind regards,

Nishith Prakash, Ph.D.

Academic Editor

PLOS ONE

Journal Requirements:

Reviewers' comments:

Reviewer's Responses to Questions

**Comments to the Author**

Reviewer #1: All comments have been addressed

Reviewer #2: All comments have been addressed

2. Is the manuscript technically sound, and do the data support the conclusions?

Reviewer #1: Yes

Reviewer #2: Yes

3. Has the statistical analysis been performed appropriately and rigorously?

Reviewer #1: Yes

Reviewer #2: Yes

4. Have the authors made all data underlying the findings in their manuscript fully available?

Reviewer #1: No

Reviewer #2: Yes

5. Is the manuscript presented in an intelligible fashion and written in standard English?

Reviewer #1: Yes

Reviewer #2: (No Response)

Reviewer #1: Comments on “Perception of rural adolescents and parents regarding child marriage: Findings of a community-based cross-sectional study in Bangladesh”

Comments

The manuscript has clarified the issues raised before. I have some specific suggestions, many of which are editorial in nature.

1. In Table 3, the agreement and disagreement always add to 100 percent in each row (as they should). Do we need both columns then? How about only agreement and showing the differences between the two columns? This suggestion applies to Table 5 as well.

2. In Table 4, the columns for parents and adolescents identical. Is there a mistake?

3. Figure 3 does not bode well aesthetically. Why use 3D bars? Please consider making it simpler, preferably making it in greyscale to make it printer friendly.

4. The authors should check the grammar and spelling of the manuscript more carefully. Is “Tripping Point Intervention” correct? I believe it is “Tipping”, not “Tripping.”

5. The authors should add an AI statement to the manuscript.

Reviewer #2: Here are a few suggestions the authors could consider working on:

1. Stick to the same terminology throughout the paper: change “positive perception” usages in the sampling and sample size section (line 250-251, for example).

2. P1 and P2 seem to be calculated ex-post and therefore cannot be used as sample size calculation parameters. If supported by other evidences, please discuss the same. If not, the sample size calculation can be moved to the appendix and presented as ex-post power calculation rather than ex-ante.

3. Remove unsubstantiated claims in the discussion section: For example, lines 476-478 are not supported with adequate citations.

4. Highlight the differences in perception among adolescents and parents in the environment domain. If youngsters support for CM cannot be fully explained, discuss the puzzle and propose what policy can do to help in this case.

**Do you want your identity to be public for this peer review?** For information about this choice, including consent withdrawal, please see our Privacy Policy

Reviewer #1: No

Reviewer #2: No

---

## [Author Response · Author response to Decision Letter 3]

4 Jul 2025

Journal Requirements:

Reference no 43 has been moved to 14

Responses to Comments of Reviewer #1

Reviewer #1: Comments on “Perception of rural adolescents and parents regarding child marriage: Findings of a community-based cross-sectional study in Bangladesh”

Comment 1. In Table 3, the agreement and disagreement always add to 100 percent in each row (as they should). Do we need both columns then? How about only agreement and showing the differences between the two columns? This suggestion applies to Table 5 as well.

Response: Thank you very much for your valuable comment. Tables 3 and 5 have been modified according to your recommendation.

Comment 2. In Table 4, the columns for parents and adolescents are identical. Is there a mistake?

Response: Thank you for the crucial query. The domain-specific median (IQR) score for parents and adolescents was the same.

Comment 3. Figure 3 does not bode well aesthetically. Why use 3D bars? Please consider making it simpler, preferably making it in greyscale to make it printer-friendly.

Response: Thanks a lot for your precise comment. Figure 2 has been constructed in greyscale as 2D bars

Comment 4. The authors should check the grammar and spelling of the manuscript more carefully. Is “Tripping Point Intervention” correct? I believe it is “Tipping”, not “Tripping.”

Response: Thank you for the in-depth observation. The spelling mistake is corrected in line 731, page 35

Comment 5. The authors should add an AI statement to the manuscript.

Response: Thanks a lot for your time-bound observation. As per your learned recommendation, the AI statement has been added.

Responses to Comments of Reviewer #2

Comment 1. Stick to the same terminology throughout the paper: change “positive perception” usages in the sampling and sample size section (line 250-251, for example).

Response: Thank you for the learned opinion. The term “positive perception” has been changed to “supportive perception” in the specific areas.

Comment 2. P1 and P2 seem to be calculated ex-post and therefore cannot be used as sample size calculation parameters. If supported by other evidence, please discuss the same. If not, the sample size calculation can be moved to the appendix and presented as an ex-post power calculation rather than an ex-ante.

Response: Thank you for your relevant comment. The reference to p1 and p2 values is provided correctly [25] at page no 10, line 251.

Comment 3. Remove unsubstantiated claims in the discussion section: For example, lines 476-478 are not supported with adequate citations.

Response: Lines 476-478 have been removed from the manuscript.

Comment 4. Highlight the differences in perception among adolescents and parents in the environment domain. If youngsters' support for CM cannot be fully explained, discuss the puzzle and propose what policy can do to help in this case.

Response: Thank you for your precise comment. On page 32, lines 641-643, it is mentioned that, unlike other domains, both participant groups showed a significant difference in perception in all items of the environmental domains. Notably, adolescents did not show significantly higher support for CM in all items of the environmental domain, but rather in some items, parents showed significantly higher support. For instance, a significantly higher proportion of parents supported CM marriage in items like social insecurity, negative attitude of the society, premarital relationship, etc. Adolescents showed higher support in items like legal and familial issues associated with CM, and a possible explanation could be a lack of practical life experience of such consequences. Appropriate interventions to inform these harmful consequences of CM using academic and mass media platforms are recommended [Page: 32, line:655-660]

While revising your submission, please upload your figure files to the Preflight Analysis and Conversion Engine (PACE) digital diagnostic tool, https://pacev2.apexcovantage.com/. PACE helps ensure that figures meet PLOS requirements. To use PACE, you must first register as a user. Registration is free. Then, log in and navigate to the UPLOAD tab, where you will find detailed instructions on how to use the tool. If you encounter any issues or have any questions when using PACE, please email PLOS at figures@plos.org. Please note that Supporting Information files do not need this step.

Response: Thank you very much for the valuable suggestion. Figure files are uploaded to the Preflight Analysis and Conversion Engine (PACE) digital diagnostic.

---

## [Editor Report · Decision Letter 3]

20 Jul 2025

Perception of rural adolescents and parents regarding child marriage: Findings of a community-based cross-sectional study in Bangladesh

PONE-D-24-18904R3

Dear Dr. Islam,

We’re pleased to inform you that your manuscript has been judged scientifically suitable for publication and will be formally accepted for publication once it meets all outstanding technical requirements.

Kind regards,

Nishith Prakash, Ph.D.

Academic Editor

PLOS ONE
---

## [Editor Report · Acceptance letter]

PONE-D-24-18904R3

PLOS ONE

Dear Dr. Islam,

I'm pleased to inform you that your manuscript has been deemed suitable for publication in PLOS ONE. Congratulations! Your manuscript is now being handed over to our production team.

Kind regards,

on behalf of

Professor Nishith Prakash

Academic Editor

PLOS ONE